# Korean Version of the Nursing Student Attitudes and Knowledge toward Lesbian, Gay, Bisexual, and Transgender Patients Scale

**DOI:** 10.3390/healthcare11142028

**Published:** 2023-07-14

**Authors:** Hye-Young Min, Jungmin Lee, James Montegrico, Hee-Jung Jang

**Affiliations:** 1College of Nursing, Ewha Womans University, Seoul 03760, Republic of Korea; hyeyoungmin@ewhain.net; 2School of Nursing, Hallym University, Chun-cheon 24252, Republic of Korea; j_lee0624@hallym.ac.kr; 3School of Nursing, University of North Carolina at Charlotte, Charlotte, NC 28223, USA; jmontegr@uncc.edu

**Keywords:** attitudes, knowledge, LGBT, nursing students, sexual orientation, quality of care

## Abstract

Aim: This study aimed to analyze the reliability and validity of a Korean version of the Nursing Student Attitudes and Knowledge Toward Lesbian, Gay, Bisexual, and Transgender Patients (K-NAKL) Scale, which measures health and heterosexual attitudes toward LGBT individuals. Background: Lesbian, gay, bisexual, and transgender (LGBT) individuals often face discrimination and a lack of care experience on the part of healthcare professionals. Introduction: In South Korea, the current knowledge and attitude measurement tools for medical staff regarding LGBT individuals are limited, as they only focus on homosexuality and do not account for different sexual orientations. Methods: The participants were 217 nursing college students aged 18–25. The item–total correlations method and Cronbach’s alpha coefficient were used to analyze internal consistency reliability. Face validity, content validity, construct validity, and criterion validity testing were conducted to establish scale validity. We made sure to follow STROBE guidelines when carrying out this research. Results: The K-NAKL is a culturally appropriate instrument used to measure the attitudes and knowledge of Korean nursing students when it comes to LGBT health. Discussion: As LGBT health is increasingly gaining social interest, the nursing education curriculum needs to produce culturally competent graduates to meet the health needs of this vulnerable and marginalized population. The current study contributes to that goal. Conclusion: The K-NAKL is a valid and reliable tool with which to measure attitudes and knowledge regarding LGBT health among Korean nursing students. Implications for nursing: The K-NAKL can enable Korean nursing students to increase their knowledge and improve their attitudes when caring for the LGBT population. Implications for nursing policy and health policy: The study highlights the importance of incorporating LGBT-related health education into nursing curricula and developing inclusive policies to improve the quality of care and health outcomes for LGBT individuals.

## 1. Introduction

In the United States, more than 10 million people are reported to be lesbian, gay, bisexual, and transgender (LGBT), among various other gender identities [1]. In Korea, there is no official count because non-heterosexual orientations are kept private due to Confucian social concepts [2,3]. Korean national health survey studies and medical treatment records do not collect information related to sexual orientation, and thus there is no known study concerning the overall health status of LGBT people [3]. However, Kang’s study [4] estimates that roughly 1 to 5 million make up the LGBT population in Korea.

Due to their sexual orientation, gender identity, or gender expression, sexual minority groups are exposed to societal prejudice, discrimination, violence, and a lack of psychological support [5]. The LGBT community is ostracized from interpersonal relationships and has great anxiety when using medical institutions due to fear of discrimination [6]. Such fear of discrimination, stress, and prejudice from medical staff against LGBT patients leads to further health disparities [7]. Nurses who are unprepared to care for LGBT patients are less likely to respond appropriately to their health needs, protect their human rights, and provide unbiased care [6]. Therefore, nurses should be aware that LGBT patients may present with mistrust or fear of the healthcare system because they are members of socially marginalized groups [8,9]. LGBT patients are classified as vulnerable groups, similar to migrant women. In line with the trend of increasing social interest in supporting the LGBT population, medical personnel, including nurses, ought to focus their attention on the healthcare needs of this population [10]. LGBT people face challenges when receiving healthcare services. Some of these difficulties are caused by prejudices, negative discrimination, insufficient knowledge, communication issues, and a lack of care experience among healthcare professionals [11]. Research on the LGBT population extensively highlights health disparities and discrimination in healthcare, while at the same time addressing social inequality, such as social prejudice and human rights protection. Therefore, there is a pressing need for acceptance-based and unbiased nursing for LGBT patients, as well as comprehensive training on LGBT knowledge and attitudes for medical staff, including nurses.

Existing research has brought attention to the pervasive discrimination faced by members of the LGBT community in healthcare settings. Numerous studies have documented instances of healthcare disparities, including inadequate access to care, lower quality of care, and negative experiences with healthcare providers. LGBT individuals often encounter biased attitudes, lack of understanding, and stigmatization when seeking medical treatment. These discriminatory practices can result in delayed or suboptimal care, leading to adverse health outcomes. Despite the growing recognition of these issues, there remains a pressing need for additional studies to further investigate and shed light on the specific challenges faced by LGBT individuals in healthcare settings. These studies can provide crucial insights into the nature and extent of discrimination, inform the development of effective interventions, and contribute to the overall improvement of healthcare experiences for the LGBT community.

However, the knowledge and attitude measurement tool for LGBT people currently used in South Korea is the Knowledge and Attitudes Tool [12,13] for homosexuality, including lesbians and homosexuals. This is limited when it comes to measuring knowledge of, and attitudes toward, different sexual orientations, which refers to the diverse range of patterns of emotional, romantic, and sexual attractions that individuals may experience. Thus, it is important to understand and assess individuals’ awareness, understanding, and acceptance of various sexual orientations, regardless of their own gender identity.

Therefore, the current study aimed to confirm the reliability and validity of a tool developed by Cornelius and Carrick [14] designed to gauge nursing students’ knowledge of, and attitudes toward, LGBT healthcare issues. Indeed, once it has been translated into Korean and demonstrated good reliability and validity, the tool could be utilized as a health knowledge and heterosexual attitude measurement instrument for Korean nursing students in LGBT health education programs. In the future, we will objectively evaluate the level of knowledge and heterosexual attitudes of Korean nursing college students using the scale cross-culturally translated and adapted in this study, accumulating basic data for the development of LGBT nursing education programs in South Korea.

## 2. Materials and Methods

### 2.1. Design

This was a methodological study that sought to evaluate the validity and reliability of a Korean version of the Nursing Student Attitudes and Knowledge Toward Lesbian, Gay, Bisexual, and Transgender Patients (K-NAKL) Scale.

### 2.2. Ethical Considerations

The study was conducted according to the guidelines of the Declaration of Helsinki and approved by the Institutional Review Board of Hallym University (HIRB-2022-029). Participants were notified that (1) the survey would be anonymous, (2) they had the right to withdraw from the study at any time without any disadvantage, and (3) their privacy and confidentiality would be guaranteed. Informed consent was obtained from all participants before beginning the study.

### 2.3. Participants

The participants were 217 Korean nursing college students aged 18 to 25 who understood the purpose of this work and agreed to participate. The sample size was based on past work, which concluded that more than 200 samples are needed to obtain reliable factors for exploratory factor analysis (EFA) [15]. The Strengthening the Reporting of Observational Studies in Epidemiology (STROBE) guidelines were used for reporting the findings of this study.

### 2.4. Measurement

#### 2.4.1. The LGBT Health Concern Knowledge and Attitude Measurement Tool

Describing data as a development process can be misleading. The instruments used for assessing reliability and validity in this study were applied cross-culturally and were originally developed by Cornelius and Carrick [14]. The Nursing Students’ Knowledge of and Attitudes Toward LGBT Healthcare [14] was developed to measure the knowledge that nursing college students possess regarding various areas of LGBT individual healthcare (e.g., cancer risk, cardiovascular risk, vaccination, human immunodeficiency virus infection and acquired immune deficiency syndrome, sexually transmitted diseases, drug abuse, sex reassignment surgery, smoking, and violence). This tool consists of 37 items, measuring knowledge and attitudes separately. Knowledge was evaluated using three categories: “true”, “false”, and “I don’t know”. The correct answers were given 1 point, while incorrect and “I don’t know” answers were given 0 points. However, whilst Kuder–Richardson 20 (KR-20) was not mentioned in the original study, the KR-20 for the adapted 34 items in this study was 0.87.

Attitudes were measured using a five-point Likert scale in response to 26 questions, which were developed to evaluate the level of comfort and attitudes of nursing students when it came to providing care for LGBT patients. The score ranged from strongly agree (1 point) to strongly disagree (5 points), meaning that the higher the average score, the higher the comfort level of the participant in providing LGBT-related care. In Cornelius and Carrick’s [14] study, Cronbach’s alpha was 0.77, and it was 0.94 for the 21 adapted items in this study.

#### 2.4.2. General Knowledge of Providing Care to LGBT Patients

The Knowledge About Homosexuality Questionnaire was originally developed by Harris [16] and cross-culturally adapted by Kang and Min [3]. This tool consists of 15 binary questions of true and false categories. If the answer was correct, 1 point was allocated, and if participants answered incorrectly, a score of 0 points was given. The higher the score, the greater the participant’s knowledge regarding providing care to LGBT patients. The KR-20 in Kang and Min’s [3] study was 0.58, and 0.43 in the current study.

#### 2.4.3. General Attitudes toward LG Patients

The general attitude toward lesbian and gay (LG) patients was measured using the Attitude Toward Lesbians and Gay Men (ATLG) Scale, which was cross-culturally adapted by Yoon et al. [13]. The ATLG Scale comprises 20 questions rated on a five-point Likert scale to measure attitudes toward gay and lesbian men and women, such as the perception of sexual taste, discrimination, and prejudice. The higher the score, the more prejudice against, and misconceptions about, LGBT individuals there are among participants. Cronbach’s alpha was 0.90 in the original study [13] and 0.94 in the adapted study. In the present study, Cronbach’s alpha was 0.68.

### 2.5. Study Process

#### 2.5.1. Translation Process

We followed the cross-cultural adaptation guidelines outlined by the World Health Organization [17] for our translation process, which included semantic translation. Initially, two independent translators, both native Korean speakers with proficiency in English, carried out the forward translation. The co-author was involved in the forward translation, along with a professor from the School of Nursing in South Korea, who were not affiliated with our study, participated in the forward translation process. Both forward translators possessed degrees or had been visiting scholars in the USA, ensuring their linguistic competence.

Following the forward translation, an expert panel conducted a comprehensive review to identify and resolve any inconsistencies between the original English version and the translated Korean version. This panel consisted of three experts, including the two experts from the forward translation phase. Another panel member was a professor from the School of Nursing in South Korea, who is an expert in risky sexual behavior.

Subsequently, two independent translators, fluent in both English and Korean and currently employed as nurses in the USA, conducted backward translation of the tool. To ensure the quality of the translation, all five experts collectively compared and deliberated on the English and Korean versions, resulting in the development of the pre-final Korean version of the tool.

Finally, four experts (excluding the co-author) rated each item on the Korean version, leading to the calculation of a content validity index (CVI) of 1.0, indicating unanimous consent agreement on all items within the Korean version of the NKALH.

#### 2.5.2. The K-NAKL Scale Validation Process

##### Content Validity

The content validity for the K-NAKL Scale was examined by three assistant professors and two professors currently teaching at the nursing college. As Polit et al. [18] suggested, the item-level content validity index (I-CVI) was used to evaluate content validity. In the current study, all 37 items had an I-CVI of 0.80 or higher. Thus, all of them were selected without being eliminated. In this process, sociocultural factors were also taken into account.

##### Face Validity

A pilot test was conducted to verify the face validity of the K-NAKL Scale. We recruited six junior students, and they reported that there were no difficulties when it came to the readability and comprehensibility of the questionnaire. Since each student’s understanding level varies, face validity was measured using junior students rather than senior students.

##### Construct Validity

EFA was used to examine the construct validity of the K-NAKL Scale with varimax rotation.

##### Criterion Validity

We compared the K-NAKL Scale with the existing tool used to measure general knowledge of, and attitudes toward, LGBT patients.

#### 2.5.3. The Internal Consistency Reliability of the K-NAKL Scale

The item–total correlation (ITC) method and Cronbach’s alpha were used to verify the reliability of the K-NAKL Scale.

### 2.6. Data Collection

The data collection was conducted through social networking services from August 2022 to January 2023. If the participants agreed to participate in the study after reading the information sheet, they were asked to click “agree” at the bottom of the first page and proceed to the survey. A total of 244 participants submitted the survey; however, due to incomplete surveys and meeting the exclusion criteria, only 217 samples were ultimately used for the final analysis.

### 2.7. Data Analysis

The collected data were analyzed using the SPSS/WIN 25.0 and R programs [19].

## 3. Results

### 3.1. General Characteristics of the Participants (n = 217)

Table 1 presents the general characteristics of the participants. Most participants were heterosexual (88.5%, n = 192) and female students (84.3%, n = 183). The average age of the students was 21.06 (±1.728) years. The proportions of percentages were similar across grades. Approximately 7 out of 10 participants did not follow any religion (65.9%, n = 143). Excluding themselves, 75.6% (n = 164) of the students had LGBT friends and/or acquaintances, and 62.3% had a normal-to-high interest in the LGBT population (62.3%, n = 135). One in three (60.4%, n = 131) participants did not believe that the LGBT population should be socially protected. Approximately 80% of students had not received any education on caring for LGBT patients (78.3%, n = 170), and the most frequently used teaching methods were face-to-face (10.1%, n = 22) and online education (7.8%, n = 17). More than 70% of students thought that education on nursing care for LGBT patients was needed (71%, n = 154), with the most preferred teaching method being face-to-face education (36.4%, n = 79), followed by online education (28.1%, n = 61), and discussion or seminars (15.7%, n = 34).

### 3.2. Validity and Reliability of the Adapted Instrument concerning Knowledge of Nursing Care for LGBT Patients

This tool measures nursing students’ knowledge of, and attitudes toward, LGBT patients. We used different methods to verify the validity of the knowledge and attitude measures because students’ knowledge was scored dichotomously (correct and incorrect answers), whereas attitude was scored on a five-point Likert scale.

The measure of students’ knowledge regarding LGBT patients’ needs comprised 37 questions, and the average item-level content validity (I-CVI) score was 0.82 [20]. An item analysis of the students’ knowledge is shown in Table 2. The item difficulty index (IDI) and discrimination index (DCI) were used to evaluate the construct validity of the tool. The IDI and DCI are based on the classical test theory (CTT) and item response theory (IRT).

For item discrimination, when CTT was applied, items with a DCI of 0.40 or higher were seen as having high discriminative power, those with a DCI of between 0.21 and 0.40 were classed as having acceptable discriminative power, and those with a DCI of between 0.10 and 0.20 were deemed to have somewhat low discriminative power. An item with a value less than 0.10 may be considered to have no discrimination. In addition, when IRT was applied, items with a DCI of 1.70 or higher (10) were classed as very high discrimination items, those with 1.36 to less than 1.69 (6) as high discrimination items, 0.68 to 1.35 (9) as appropriate discrimination items, 0.34 to 0.67 (7) as low discrimination items, and 0.33 to 0.10 (2) as very low discrimination items.

The DCI applied with CTT ranged from 0.04 to 0.58, with an average of 0.29. Eight items (21.6%) had high discriminative power, 16 items (43.2%) had moderate levels of discriminative power, and 13 items (35.2%) had low discriminative power. Three items with a value of less than 0.10 were judged to have no DCI and were removed (items 2, 21, and 36). The removed items were “Lesbians are more likely to suffer from obesity than heterosexual women” (#2), “Sex reassignment surgery is being performed all over the world” (#21), and “Male-to-female transgender people are more vulnerable to suicide than the other way around” (#36). Subsequently, the analysis was conducted using 34 questions, with three questions removed.

When IRT was applied, there were no items with negative values, and the average discrimination power index value was 1.25, which was an appropriate level. Eleven items (32.4%) had very high discriminative power, five items (14.7%) showed high discrimination, nine items (26.5%) exhibited appropriate discrimination, seven items (20.6%) showed somewhat low discrimination, and two items (5.8%) had low discriminative power. Item 22, “Gay men and lesbians who do not disclose their sexual orientation are at increased risk of developing melanoma and other cancers due to psychogenic immunosuppressive reactions”, had the lowest discrimination level of 0.31. Question 16 also had a low score of 0.33, with low discriminative power (“LGBT people are less likely to be in long-term monogamous relationships than non-LGBT people”). By contrast, item 17, “Female-to-male (transgender) people with a cervix are at risk of cervical cancer and need regular cervical cytology”, recorded a very high discrimination score of 2.49.

For the IDI, when CTT was applied, scores of 0.80 or higher were interpreted as easy, scores between 0.30 and 0.80 as appropriate, and scores of less than 0.30 as difficult. When IRT was applied, the score generally ranged between −2.0 and 2.0, where −2.0 or less was very easy, −2.0 to −0.5 was easy, −0.5 to 0.5 was medium, 0.5 to 2.0 was difficult, and 2.0 or more was very difficult.

The results for IDI applying CTT showed that 13 (38.2%) items had appropriate difficulty, and 21 (61.8%) items were difficult. There were no easy items. Applying IRT illustrated that each item was evenly distributed from easy to very difficult. Items that were considered difficult were the most common (35.3%, n = 12), followed by those with easy and medium levels of difficulty (23.5%, n = 8, each) and very difficult (17.6%, n = 6) items. None of the items were considered very easy. The most difficult item was “When asking adolescents about sex, it is important to first ask about their sexual behavior rather than their gender identity” (#8), with an IDI of 4.42. The easiest item was “Lesbians do not need Pap tests as often as heterosexual women” (#14), with an IDI of −1.04.

The ITCs for 34 items were considered somewhat acceptable, and the alpha of the item-deleted value, which indicates the change in Cronbach’s alpha value when measurement items are removed, did not change significantly. Therefore, we retained all 34 items (KR-20 was 0.87). The result seen after measuring the correlation between the two scales showed no statistical significance (r = 0.059, *p* = 0.388).

### 3.3. Factor Analysis of the Adapted Instrument for Attitudes toward Nursing Care for LGBT Patients

The measure of students’ attitudes toward LGBT patients comprised 26 questions, and the average I-CVI was 0.90. The factor analysis results are presented in Table 3. The Kaiser–Meyer–Olkin (KMO) value for factor analysis adequacy was 0.923, and the results of Bartlett’s sphericity test were also significant (χ^2^ = 3445.117, *df* = 210, and *p* < 0.001), thus confirming that the data were adequate for factor analysis. In the results of the first EFA with varimax rotation, item 13 did not belong to any factor and was removed. When the second EFA was run, item 6 was removed because it loaded onto two factors. When EFA was run again, item 2 did not load onto any factor; therefore, it was removed. When the fourth EFA was run, only one item (#1) was loaded onto a factor, and so it was removed. Finally, when EFA was run for 22 items, item 23, with a factor loading of less than 0.05, was removed. The factor loadings of 21 items all exceeded 0.4 (0.51–0.87). Principal component analysis revealed three factors with eigenvalues of 1 or more, and the cumulative variance explained by the three factors was 65.14%. Cronbach’s alpha values for each of the subscales or factors were 0.91 (Factor 1), 0.90 (Factor 2), and 0.84 (Factor 3), which reflected the adequate reliability of each of the subscales.

Skewness and kurtosis were checked to confirm the normal distribution of scores based on the criteria of Kline [21]. Skewness did not exceed an absolute value of 3, and the kurtosis did not exceed an absolute value of 8, indicating a normal distribution. The ITC ranged from 0.26 to 0.85, and the average was 0.64. Based on the ITC, among the 21 items, one item (#4) had a low discrimination level of less than 0.3. However, there was no significant effect on internal consistency, even when this item was removed. Thus, the final 21 items were maintained without removing item 4.

The first and second factors contained nine items each. The first factor could be labeled “Absence of empathy for people of different sexual orientations” (eigenvalue, 10.25; variance explained, 48.80%), and the second factor could be named “Discrepancy when providing care despite acknowledging a different sexual orientation” (eigenvalue, 2.06; variance explained, 9.82%). Finally, the third factor may be interpreted as “Fully understanding others’ sexual orientation” (eigenvalue, 1.37; variance explained, 6.52%).

As a result of verifying the correlation between the adapted and existing tool for criterion validity verification using the Pearson correlation coefficient, we found a moderate to strong negative correlation (r = −0.652, *p* < 0.001). Cronbach’s alpha for the 21 items was 0.94.

## 4. Discussion

The adaptation of the NKALH Scale adds to the literature by providing a culturally appropriate instrument for measuring the attitudes and knowledge of Korean nursing students regarding LGBT individuals. The ITC method and Cronbach’s alpha coefficient were used to analyze the internal consistency reliability of the said scale. To establish the validity of the adapted instrument, face validity, content validity, construct validity, and criterion validity testing were conducted [22]. The limited number of existing Korean versions of instruments that measure knowledge of, and attitudes toward, LGBT health [11] was the main impetus for this study. The nursing education curriculum needs to produce culturally competent graduates in order to meet the health needs of this vulnerable and marginalized population [3,23]. The development of the K-NAKL Scale adds to the literature by providing a culturally appropriate instrument for measuring the attitudes and knowledge of Korean nursing students regarding LGBT individuals.

The findings show that the K-NAKL Scale is a valid and reliable tool with which to assess the attitudes and knowledge of Korean nursing students when it comes to LGBT-related health. The I-CVI of the K-NAKL Scale was 0.82, which is an acceptable measure of content validity [20]. To determine the construct validity, five iterations of EFA were undertaken, with 21 items adapted, and three factors were identified using simple structure guidelines. The Pearson correlation coefficients between the existing and adapted tools yielded a statistically significant result (*p* < 0.001), which established the criterion validity of the K-NAKL Scale. ITC and Cronbach’s alpha coefficient provided adequate evidence to establish the internal consistency reliability of the instrument [24,25].

### 4.1. Participants’ Characteristics

More than half (51.7%) of the participants were in their freshman and sophomore years of nursing education, and they may not have received proper training on LGBT health at this stage. Most participants (78%) reported that they had not received education on caring for LGBT individuals. Third, less than 1 out of 10 participants (8.8%) believed that the LGBT population should be socially protected.

### 4.2. Validity of the K-NAKL Scale

#### 4.2.1. Content Validity

Face validity and I-CVI were used to assess the content validity of the K-NAKL Scale. Content validity is essential when it comes to analyzing the representativeness of the domains (i.e., knowledge of, and attitudes toward, LBGT-related health) [26]. In this study, face validity testing was conducted in a pilot study to determine the readability and comprehensibility of the instrument items. While face validity does not reflect a true measure of validity [25], it does provide information regarding the usability of the tool from prospective participants. An I-CVI of 0.82 for all knowledge items and 0.90 for attitude items from five nurse educators reflected the adequacy and importance of the items in measuring the domains on the K-NAKL Scale. An acceptable I-CVI indicates a high level of agreement from the subject matter experts on the items in an instrument [27].

#### 4.2.2. Construct Validity

Instruments that support existing theoretical constructs are said to have good construct validity [25]. Using CTT and IRT, the IDI and DCI were employed to evaluate the construct validity of the K-NAKL Scale. EFA with varimax rotation was utilized to determine the factorability of the items; the KMO, Bartlett’s test of sphericity, and ITC were used to analyze the adequacy of the EFA [28].

The IDI showed that most items were difficult using CTT, but when IRT was applied, the items were distributed from easy to very difficult, with difficult items considered the most common. The level of difficulty of most items can be attributed to two factors: more than half of the participants in this study were in their freshman and sophomore years of nursing education, and most of them reported not receiving any education on LGBT-related health. The DCI had means of 0.29 and 1.25 applying CTT and IRT, respectively, which indicate appropriate levels of construct validity. DCIs that are 0.20 and above indicate an adequate norm-referenced measure [25].

EFA was iteratively used five times until the results satisfied simple structure guidelines. The EFA yielded three factors with 21 items (out of the original 26 items), which were included due to the fact that they had a factor loading of at least 0.40, and each item was loaded onto only one factor. The ITC further supported the factorability of the 34 items that were retained. When measurement items were removed, it was noted that the alpha of the item-deleted value did not significantly change the alpha coefficient (KR-20 = 0.87) of the overall scale. Additionally, the alpha coefficient of each of the subscales or factors (KR-20 = 0.84 to 0.91) strongly supported the reliability and multidimensionality of the instrument. KR-20 was used, as it is the appropriate alpha version for dichotomous items [15].

To assess the appropriateness of using EFA, skewness and kurtosis were determined, and they both confirmed the normal distribution of scores. The KMO (0.923), Bartlett’s sphericity test (*p* = < 0.001), and ITC (0.87) results all supported the adequacy of the factor analysis [25]. Moreover, principal component analysis confirmed three factors based on eigenvalues of 1 or more, and these factors accounted for 65.1% of the cumulative variance explained by the K-NAKL Scale.

Interestingly, the three factors in the K-NAKL Scale, namely “Absence of empathy for people of different sexual orientations” (Factor 1), “Discrepancy when providing care despite acknowledging a different sexual orientation” (Factor 2), and “Fully understanding others’ sexual orientation” (Factor 3), were reflective of the participants’ characteristics and responses. Factor 1 accounted for 48.8% of the variance in the K-NAKL Scale. This may be a reflection of the belief of most participants that the LGBT population should not be socially protected. Only 8.8% believed that LGBT individuals ought to be socially protected. This can be explained by the Confucian social concepts in Korea regarding the LGBT population [3]. Factor 2 may be related to the variations in the level of nursing education of the participants, where more than half were still in the freshman and sophomore years of their nursing program. At this stage in nursing education, students at a lower level may not have received training on certain items of the instrument that measured knowledge regarding LGBT-related health. Additionally, most participants reported that they had indeed not received any education on caring for LGBT individuals but believed that education regarding nursing care for this vulnerable population is needed [7].

#### 4.2.3. Criterion Validity

The criterion validity of the K-NAKL Scale was assessed by determining the correlation coefficients between the adapted and existing tools using the Pearson correlation. There was a significant moderate to strong negative correlation between the tools, which suggests that the adapted and existing tools measure the same domains [25].

### 4.3. Reliability of the K-NAKL Scale

ITC and Cronbach’s alpha were used to determine the internal consistency reliability of the K-NAKL Scale. ITC measures the magnitude of the relationship between the items of a scale and the total score of the scale, wherein ITC scores greater than 0.20 are generally acceptable [29]. The mean ITC of the adapted attitudes scale was 0.64. While one item had a low discrimination level, removing the said item did not significantly impact the internal consistency; thus, all 21 items were retained. Moreover, the mean ITC of the adapted knowledge scale was 0.37.

The reliability coefficient of the original tool, which measured knowledge, was not reported. In this study, the reliability coefficient was adequate (0.87). Conversely, the reliability coefficient of the original tool that measured attitudes was 0.77, which was lower than that in the current study (0.94). A reliability coefficient of 0.70 is used as the minimum acceptable level of reliability of a measurement instrument [25].

### 4.4. Implications for Nursing and Health Policy

This study emphasizes the need for nursing education programs to incorporate comprehensive training on LGBT health. By integrating LGBT-related content into the curricula, nursing students can develop the necessary knowledge and skills needed to provide culturally competent care to LGBT individuals. This inclusion will contribute to reducing discrimination, communication barriers, and the lack of care experience that LGBT individuals often face in healthcare settings.

The findings of this study highlight the importance of developing inclusive policies within healthcare institutions and organizations. These policies should promote a welcoming and affirming environment for LGBT individuals, address discriminatory practices, and ensure equal access to healthcare services. Implementing such policies will help create a supportive climate for LGBT patients and foster better health outcomes by reducing barriers to care.

By implementing the K-NAKL Scale and promoting its use among nursing students, healthcare institutions can enhance the quality of care provided to LGBT individuals. The scale can serve as a tool with which to assess and monitor nursing students’ attitudes and knowledge regarding LGBT health, allowing for targeted interventions and educational initiatives to address any knowledge gaps or negative attitudes. This will ultimately lead to improved health outcomes and a more inclusive healthcare system. The present study offers an essential contribution to the field, shedding light on the knowledge and attitudes of nursing students when it comes to LGBT healthcare experiences. It provides a foundation for further research, highlighting the need for inclusive education and training in nursing programs, as well as the development of culturally sensitive healthcare practices to better support the LGBT community.

The study highlights the need for nursing policy and health policy to prioritize cultural competence training for healthcare professionals. By fostering an understanding of diverse sexual orientations and gender identities, healthcare providers can deliver respectful and patient-centered care to LGBT individuals. This cultural competence training should be an integral part of continuing education programs, licensure requirements, and professional development opportunities for nurses. Incorporating LGBT-related health education and developing inclusive policies can contribute to reducing health disparities among the LGBT population. Healthcare and nursing policies should prioritize addressing the specific health needs of LGBT individuals, including mental health, sexual health, and access to gender-affirming care. By promoting equity and eliminating barriers to care, nursing and health policies can help ensure that LGBT individuals receive the same standard of care as their heterosexual counterparts.

Having this perspective acknowledges that individuals and healthcare systems must continuously learn, adapt, and improve their understanding of, and responsiveness to, diverse cultural backgrounds, including the needs of the LGBT community. Furthermore, it is important to acknowledge that any assessment tool, including the K-NAKL Scale, has the potential to perpetuate stereotypes and biases through the inclusion of certain items. It is crucial for researchers and developers of such tools to exercise caution and critically evaluate the items included to ensure that they do not reinforce harmful assumptions or perpetuate discrimination. In light of this, it is necessary for the authors to acknowledge the limitations and potential biases of the K-NAKL Scale tool in their study.

By recognizing the need to move beyond cultural competence and acknowledging the potential biases of assessment tools, nursing practice can shift toward a more inclusive and nuanced approach. This involves embracing cultural effectiveness or humility, and emphasizing the continuous learning and growth required to provide culturally sensitive care to diverse populations, including the LGBT community. Moreover, it highlights the importance of critically evaluating and refining assessment tools to ensure that they align with the principles of inclusivity, respect, and equity in healthcare practice.

Overall, analyzing the reliability and validity of the K-NAKL Scale is crucial for advancing our understanding of LGBT healthcare experiences. By assessing the psychometric properties of the K-NAKL Scale, we gain a better understanding of the tool’s effectiveness in capturing the nuances of LGBT individuals’ needs and experiences in healthcare settings. This, in turn, contributes to the broader goal of improving nursing practice and patient care. The study’s implication for nursing practice lies in the potential to enhance the education and training of healthcare professionals regarding LGBT issues. A valid and reliable tool such as the K-NAKL Scale can serve as a foundation for developing targeted educational interventions that address the specific needs of LGBT patients, thereby promoting inclusive and culturally competent care. Thus, the analysis of reliability and validity directly supports the identification of gaps in nursing education and highlights the necessity of further education around LGBT issues in healthcare practice. The implications for nursing and health policy based on this study underscore the importance of fostering cultural competence, developing inclusive policies, and addressing the unique health needs of LGBT individuals. By implementing these recommendations, nursing education and healthcare systems can better serve and support the well-being of the LGBT population.

### 4.5. Limitations

Despite efforts made to address potential sources of bias, it is important to acknowledge the limitations associated with convenience sampling—the non-probability sampling method employed in this study. Convenience sampling introduces biases due to the non-random selection of participants based on their easy availability or accessibility. Given these limitations, the findings should be interpreted with caution as they may not be generalizable to the larger population of interest. The researchers have made efforts to mitigate biases by recognizing and reporting the limitations, supplementing convenience sampling with other methods, employing statistical adjustments, and conducting sensitivity analyses. However, it is crucial to acknowledge that convenience sampling inherently restricts the diversity and representativeness of the sample, limiting the extent to which the findings can be generalized beyond the study population. The K-NAKL Scale tool was originally developed to explore knowledge of, and attitudes toward, LGBT healthcare experiences among nursing students. The omission of explicit discussion on Q-related individuals is due to resource constraints, limited sample size, or the specific research questions driving the study. However, the exclusion of Q-related individuals in this study should not undermine the importance and significance of their experiences and perspectives. The ‘Q’ part of LGBTQ-related matters deserves attention and further research to understand the unique challenges and healthcare needs of individuals who identify within this category. Future studies should aim to specifically explore the experiences, knowledge about, and attitudes toward Q-related individuals among nursing students and healthcare professionals to ensure a comprehensive understanding of the diverse spectrum of LGBT experiences. By recognizing these limitations, future research can build upon this study’s findings and strive for a more comprehensive understanding of the healthcare experiences of all members within the LGBTQ+ spectrum, including Q-related individuals.

The exclusion of explicit discussion on LGBTQ-related thoughts of Korean nursing school students is a notable limitation of this study. While the study focused on exploring knowledge of, and attitudes toward, LGBT healthcare experiences among nursing students, it is important to recognize the significance of understanding the thoughts, perspectives, and experiences of LGBTQ individuals within the student population. By specifically investigating LGBTQ-related thoughts, future research can gain insights into the unique challenges, concerns, and aspirations of LGBTQ nursing students in relation to their education, training, and future practice. This understanding can inform the development of targeted interventions and support systems to create inclusive and affirming learning environments for LGBTQ students. Additionally, exploring LGBTQ-related thoughts among nursing students can contribute to a broader understanding of the factors influencing the provision of culturally sensitive care to the LGBTQ community. Therefore, future studies should aim to explicitly explore LGBTQ-related thoughts among Korean nursing school students to ensure a more comprehensive understanding of their experiences and to guide efforts in fostering inclusive nursing education and practice.

It is important to acknowledge several limitations in cross-cultural adaptation studies. Despite efforts to ensure the accurate adaptation of measures across different cultures, limitations include the challenge of fully capturing the cultural context and achieving complete linguistic equivalence. The cultural relevance and appropriateness of measures may vary across contexts, and convenience sampling methods can introduce selection bias. Additionally, the generalizability of findings to other cultural groups or contexts may be limited. Future studies should continue to refine cross-cultural adaptation processes to enhance the validity and applicability of measures across diverse cultural contexts.

### 4.6. Conclusions

The K-NAKL Scale exhibited good content, construct, and criterion validity, as well as adequate internal consistency reliability in the current study. Our findings support the use of the K-NAKL Scale as a valid and reliable tool with which to measure attitudes and knowledge regarding LGBT health among Korean nursing students. As more culturally appropriate instruments are needed to assess the healthcare needs of marginalized and vulnerable groups, such as the LGBT population, the addition of the K-NAKL Scale to the literature will enable Korean nursing students to increase their knowledge and improve their attitudes when caring for the LGBT population.

However, it must be considered that the validity and reliability measures of an adapted tool are largely dependent on the characteristics of the participants at the time of instrument development. Furthermore, the assessment of instrument validity and reliability is not a one-time measurement but rather a continuous and repeated process. Thus, future methodological studies to reassess the validity and reliability of the newly-developed K-NAKL Scale are needed.

## Figures and Tables

**Table 1 healthcare-11-02028-t001:** General characteristics of the participants (N = 217).

Categories		N (%)	M ± SD
Sex	Male	34 (15.7)	
	Female	183 (84.3)	
Sexual orientation	Heterosexual	192 (88.5)	
	Homosexual	2 (0.9)	
	Bisexual	12 (5.5)	
	Don’t know	11 (5.1)	
Age (years)	18–20	85	21.06 ± 1.728
	21–23	106	
	24–25	26	
Grade	Freshman	42 (19.4)	
	Sophomore	70 (32.3)	
	Junior	55 (25.3)	
	Senior	50 (23.0)	
Religion	Atheist	143 (65.9)	
	Christian	46 (21.2)	
	Catholic	12 (5.5)	
	Buddhist	16 (7.4)	
Do you have any LGBT friends and/or acquaintances? (Excluding yourself)	Yes	164 (75.6)	
	No	53 (24.4)	
Interest in the LGBT population	Not very interested	25 (11.5)	
	Not interested	57 (26.3)	
	Neutral	75 (34.6)	
	Interested	52 (24.0)	
	Very interested	8 (3.7)	
LGBT population should be socially protected	Absolutely no	28 (12.9)	
	Somewhat no	103 (47.5)	
	Don’t know	67 (30.9)	
	Somewhat yes	16 (7.4)	
	Absolutely yes	3 (1.4)	
Have you had LGBT nursing education experience in the past?	Yes	47 (21.7)	
	No	170 (78.3)	
If yes, what was the teaching method?	Face-to-face education (lecture)	22 (10.1)	
	Online education	17 (7.8)	
	Simulation	2 (0.9)	
	Discussion or seminar	6 (2.8)	
Need for LGBT nursing education	Yes	154 (71)	
	No	63 (29)	
Preferred teaching methods for LGBT nursing education	Face-to-face education (lecture)	79 (36.4)	
	Online education	61 (28.1)	
	Simulation	22 (10.1)	
	Discussion or seminar	34 (15.7)	
	Others	21 (9.7)	

**Table 2 healthcare-11-02028-t002:** Validity analysis of the adapted instrument for knowledge of nursing care for LGBT patients.

No.	Items	Item–Total Correlation	Alpha If Item Deleted	IDI		DCI		Correct Answer n (%)
				CTT	IRT	CTT	IRT	
1	The prevalence of cervical cancer and cervical dysplasia is known to be the same among lesbians and heterosexual women.	0.158	0.866	0.20	3.590	0.17	0.401	43 (19.8)
3	Lesbians are at lesser risk of alcohol abuse than heterosexual women.	0.339	0.862	0.59	−0.534	0.30	0.762	128 (59.0)
4	The incidence of depression is higher in older gays and lesbians than in the general population.	0.415	0.860	0.53	−0.107	0.40	1.236	115 (53.0)
5	In male-to-female sex reassignment surgery, the prostate is removed.	0.227	0.864	0.23	2.125	0.19	0.624	49 (22.6)
6	Heterosexual women are more likely to smoke than lesbian women.	0.369	0.861	0.62	−0.702	0.27	0.805	135 (62.2)
7	Breast cancer can still occur after bilateral breast reduction surgery in female-to-male sex reassignment (transgender).	0.455	0.859	0.60	−0.350	0.37	1.483	130 (59.9)
8	When asking adolescents about sex, it is important to first ask about their sexual behavior rather than their gender identity.	0.125	0.866	0.14	4.423	0.10	0.418	31 (14.3)
9	The fastest-growing population of new HIV infections is men who have sex with men.	0.414	0.860	0.56	−0.251	0.37	1.184	122 (56.2)
10	The rate of domestic violence in gay male households is similar to that of the total population.	0.240	0.864	0.29	1.922	0.18	0.474	64 (29.5)
11	Being a transgender person (transgender) means experiencing a strong incongruity between one’s birth sex and gender identity.	0.352	0.862	0.62	−0.659	0.38	0.869	135 (62.2)
12	Mortality rates from suicide among lesbians and gay men are comparable as a percentage of the total population.	0.343	0.862	0.37	0.793	0.33	0.728	81 (37.3)
13	The LGBT population has unique health risks and health needs.	0.471	0.859	0.53	−0.075	0.47	1.689	115 (53.0)
14	Lesbians do not need Pap tests as often as heterosexual women do.	0.348	0.862	0.68	−1.037	0.26	0.843	148 (68.2)
15	Sex reassignment therapy is readily available and covered by most insurance plans.	0.454	0.859	0.69	−0.830	0.40	1.223	150 (69.1)
16	LGBT people are less likely to be in long-term monogamous relationships than non-LGBT people.	0.186	0.866	0.43	0.833	0.15	0.327	94 (43.3)
17	Female-to-male (transgender) people with a cervix are at risk of cervical cancer and need regular cervical cytology.	0.618	0.855	0.57	−0.152	0.57	2.492	123 (56.7)
18	Men who have sex with men have a higher risk of contracting hepatitis A than men who have sex with women.	0.439	0.860	0.31	0.724	0.37	1.614	67 (30.9)
19	Female-to-male sex reassignment (transgender people) patients receiving androgen therapy (hormone therapy) have a higher risk of cardiovascular disease.	0.503	0.858	0.44	0.232	0.46	1.962	95 (43.8)
20	Access to health care is the same for LGBT people as for other populations.	0.217	0.865	0.21	2.236	0.15	0.648	45 (20.7)
22	Gay men and lesbians who do not disclose their sexual orientation are at increased risk of developing melanoma and other cancers due to psychogenic immunosuppressive reactions.	0.153	0.866	0.28	3.088	0.13	0.310	61 (28.1)
23	Older LGBT individuals receive less HIV-related information and HIV education.	0.353	0.862	0.35	0.714	0.32	1.068	76 (35.0/0)
24	LGBT youth are at high risk of becoming victims of violence and substance abuse.	0.529	0.857	0.49	0.072	0.50	2.092	106 (48.8)
25	Nutritional deficiencies associated with eating disorders are higher in LGBT youth than in non-LGBT youth.	0.420	0.860	0.23	1.010	0.30	1.841	49 (22.6)
26	Male-to-female (transgender) people who have undergone vaginoplasty do not need penile cancer screening.	0.167	0.865	0.18	3.762	0.15	0.417	39 (18.0)
27	The survival rate of gay men with cancer is the same as that of the general population.	0.241	0.864	0.33	1.420	0.31	0.539	71 (32.7)
28	Male-to-female (transgender) people receiving estrogen therapy (hormone therapy) have an increased risk of breast cancer.	0.510	0.858	0.41	0.327	0.50	1.861	89 (41.0)
29	LGBT patients may not disclose their sexual orientation for fear of being affected by treatment.	0.551	0.857	0.76	−0.911	0.37	2.045	166 (76.5)
30	LGBT individuals are predisposed to high blood pressure.	0.338	0.862	0.12	1.545	0.18	1.911	26 (12.0)
31	LGBT individuals have a higher risk of heart disease and stroke than the general population.	0.327	0.862	0.14	1.496	0.21	1.682	31 (14.3)
32	Male-to-female (transgender) people who retain testicles are at risk of developing testicular cancer.	0.533	0.857	0.53	−0.073	0.53	1.721	115 (53.0)
33	Adult lesbians smoke more than the heterosexual female cohort.	0.390	0.861	0.22	1.058	0.26	1.816	47 (21.7)
34	Lesbians have a higher risk of breast cancer than heterosexual women.	0.304	0.863	0.11	1.793	0.15	1.616	23 (10.6)
35	LGBT individuals are often the target of harassment and violence from heterosexual people.	0.555	0.857	0.75	−0.814	0.42	2.128	162 (74.7)
37	Male-to-female (transgender) people have a higher risk of HIV infection than female-to-male (transgender) people.	0.419	0.860	0.29	0.853	0.32	1.489	62 (28.6)

**Table 3 healthcare-11-02028-t003:** Factor analysis.

	Factor 1	Factor 2	Factor 3
3	0.079	0.595	0.137
4	0.564	−0.382	0.278
5	0.348	0.280	0.593
7	0.265	0.511	−0.066
8	−0.001	0.766	0.396
9	0.294	0.593	0.423
10	0.075	0.781	0.322
11	0.598	0.363	0.106
12	0.702	0.013	0.172
14	0.244	0.222	0.870
15	0.195	0.195	0.853
16	0.796	0.264	0.162
17	0.699	0.317	0.286
18	0.729	0.344	0.151
19	0.354	0.512	0.393
20	0.779	0.174	0.114
21	0.461	0.735	0.106
22	0.519	0.680	0.221
24	0.692	0.356	0.344
25	0.571	0.568	0.360
26	0.557	0.623	0.301
Eigenvalue	10.247	2.063	1.370
Total variance explained proportion (%)	48.797	9.823	6.522
Cumulative proportion (%)	48.797	58.621	65.143

## Data Availability

The data that support the findings of this study are available from the frist author on special request.

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
