# Peer review of "Korean Version of the Nursing Student Attitudes and Knowledge toward Lesbian, Gay, Bisexual, and Transgender Patients Scale"

_healthcare, 2023, doi:10.3390/healthcare11142028_

Round 1

Reviewer 1 Report

Dear authors,

You are addressing a very important issue regarding the need for increased knowledge for healthcare providers working with members of the LGBT community. However, the manuscript in its current form has a few shortcomings. 

Please review the following considerations:

Abstract:

-aim should be the last sentence of the introduction 

- you have transgender in your LGBT – it is not considered a sexual orientation. Please be clear with your terminology. 

- last sentence of the discussion would make more sense in the introduction, highlighting the need for education. I don’t see a connection between the results and the discussion. The discussion does not highlight the connection between the results and the aim of the study. It is a bit unclear how the aim to analyze the reliability and validity of the K-NAKL and the need for more education around LGBT issues in health care is connected. Since the study did not look at attitudes and perceptions, the authors need to make a clearer connection between the aim and conclusion/implication to nursing practice. 

Line 14: communication barriers – this is a confusing statement. I would eliminate this line in the abstract unless you provide clarifications.

Line 36 – transgender is not considered a sexual orientation. 

Line 58-61: move up to the paragraph before. It feels disjointed in its current position

Line 61-62: Authors state that research is mainly focused on social inequalities. This is incorrect. There is a huge body of research documenting health disparities and discrimination in health care. If the authors are referring to practices in Korea, this needs to be made clearer. 

Introduction: Authors start out with a very thorough description of LGBT experiences in the US and Korea. Current health disparities are being explored and validated by citing current research. Authors need to add additional studies highlighting the discrimination in healthcare experiences by members of the LGBT community. 

I see an issue with the use of terminology throughout the manuscript. Authors frequently refer to transgender as being a sexual orientation. Furthermore, the authors highlight that current tools used in Korea do not address ‘other sexual orientations.’ It would be beneficial to provide some basic definitions to strengthen credibility. 

Line 68: define different sexual orientations

Line 71- ‘could well be” – rephrase

Materials and Methods:

Design: Please provide additional detail regarding the design. 

Face validity: Provide additional reasoning for selecting 6 junior students and not senior students. How the authors arrive at the number 6 as being sufficient in establishing face validity is unclear. 

Results: 

Clearly stated. Relevant tables are included.

Discussion: Statements such as “LGBT health is increasingly gaining social interest” (line 291) can become problematic since it distracts from the fact that people of the LGBT community are not of social interest but are experiencing increasing discrimination in all areas of life. By using increased social interest as the argument why we need educated health care providers reduces the lived experience to a mere social interest group. The discussion would benefit from a reformatting to integrate validity findings, participant characteristics, and other subsections to past and current literature or tool validation practices. The authors repeat information from the results section. 

The implication section is very strong, but it remains unclear how the findings of the current study highlight the importance of developing inclusive policies. I do agree that the use of the K-NAKL can contribute to establishing a need for healthcare providers to be educated. The authors refer to cultural competence training which is not further elaborated. Current practices recommend moving away from cultural competence to cultural effectiveness or humility to denote that it is a continuous journey and not something one achieves competence. It should be mentioned that the K-NAKL tool in itself can perpetuate stereotypes and biases by the items it includes.

Conclusion:

Excellent!!

Author Response

We would like to express our appreciation for your extremely thoughtful suggestions. Your feedback was extremely helpful to strengthen our manuscript. As you will see below, we have been able to revise and improve the paper as a result of your valuable feedback.

Overall, we have made changes throughout the paper that address the points you have made as shown below. After correcting the manuscript according to the reviewers’ and editors’ comments, we got this paper revised by an academic revision company again. The corrected parts have been marked with Red Font and indicated with page numbers in the table below for easy reference.

Thank you again for taking the time to share your constructive feedback.

Yours sincerely,

Reviewer 2 Report

As an exercise in the collection of data regarding an understudied topic, this article draft succeeds in accomplishing stated goals that concern the appropriateness of the K-NAKL tool as a way of discerning LGBT-related attitudes among Korean nursing school students. We particularly appreciated the presence of an introduction that, while brief, lucidly presented the cultural context as well as some standard terminology relevant to the topic---an introductory section unfortunately lacking or barely definable in other paper drafts of this genre. The reliability of the K-NAKL tool for the purposes of this study had a careful empirical treatment (lines 297~306) and the discussion of implications also seemed properly restrained by the nature of the data---the compilers of the article clearly wanted to offer recommendations in a limited way commensurate with the reality of the deeply unexplored nature of this phenomenon (lines 427~445, 455~460). We merely wonder about one small notion not explicitly mentioned in the article, the idea of LGBTQ-related thoughts of Korean nursing school students. While the "Q" part of LBGTQ-related matters seems often barely mentioned by the mainstream media, it would seem perhaps slightly more optimal to cover these individuals (than to not cover those individuals). If the authors have declined to accommodate the discussion of Q-related people, then perhaps the authors could briefly explain the reasons for that exclusion---but otherwise, this paper presents an absolutely necessary contribution to the relevant scholarly literature.

Author Response

We would like to express our appreciation for your extremely thoughtful suggestions. Your feedback was extremely helpful to strengthen our manuscript. As you will see below, we have been able to revise and improve the paper as a result of your valuable feedback.

Overall, we have made changes throughout the paper that address the points you have made as shown below. After correcting the manuscript according to the reviewers’ and editors’ comments, we got this paper revised by an academic revision company again. The corrected parts have been marked with Red Font and indicated with page numbers in the table below for easy reference.

Thank you again for taking the time to share your constructive feedback.

Yours sincerely,

The authors
